# DISN: Deep Implicit Surface Network for High-quality Single-view 3D Reconstruction

**Weiyue Wang\*,[1]**    **Qiangeng Xu\*,[1]**    **Duygu Ceylan[2]**    **Radomir Mech[2]**    **Ulrich Neumann[1]**

[1]University of Southern California
Los Angeles, California
{weiyuewa,qiangenx,uneumann}@usc.edu

[2]Adobe
San Jose, California
{ceylan,rmech}@adobe.com

## Abstract

Reconstructing 3D shapes from single-view images has been a long-standing research problem. In this paper, we present *DISN*, a Deep Implicit Surface Network which can generate a high-quality detail-rich 3D mesh from a 2D image by predicting the underlying signed distance fields. In addition to utilizing global image features, DISN predicts the projected location for each 3D point on the 2D image and extracts local features from the image feature maps. Combining global and local features significantly improves the accuracy of the signed distance field prediction, especially for the detail-rich areas. To the best of our knowledge, DISN is the first method that constantly captures details such as holes and thin structures present in 3D shapes from single-view images. DISN achieves the state-of-the-art single-view reconstruction performance on a variety of shape categories reconstructed from both synthetic and real images. Code is available at https://github.com/laughtervv/DISN. The supplementary can be found at https://xharlie.github.io/images/neurips_2019_supp.pdf.

## 1  Introduction

Over the recent years, a multitude of single-view 3D reconstruction methods have been proposed where deep learning based methods have specifically achieved promising results. To represent 3D shapes, many of these methods utilize either voxels [2–9] or point clouds [10] due to ease of encoding them in a neural network. However, such representations are often limited in terms of resolution. A few recent methods [11–13] have explored utilizing explicit surface representations in a neural network but make the assumption of a fixed topology, limiting the flexibility of the approaches. Moreover, point- and mesh-based methods use Chamfer Distance (CD) and Earth-mover Distance (EMD) as training losses. However, these distances only provide approximated metrics for measuring shape similarity.

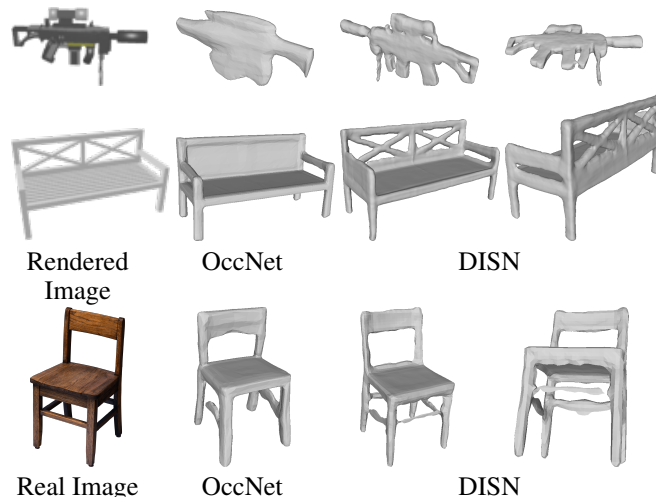

Figure 1: Single-view reconstruction results using Occ-Net [1] and DISN on synthetic and real images.

To address the aforementioned limitations in voxels, point clouds and meshes, in this paper, we study an alternative implicit 3D surface representation, Signed Distance Functions (SDF). SDFs have recently attracted attention from researchers and few other works [14–16, 1] also choose to reconstruct 3D shapes by generating an implicit field. However, such methods either generate a binary occupancy grid or consider only the global information. Therefore, while they succeed in recovering overall shape, they fail to recover fine-grained details. After exploring different forms of the implicit field and the information that preserves local details, we present an efficient, flexible, and effective Deep Implicit Surface Network (DISN) for predicting SDFs from single-view images (Figure 1).

An SDF simply encodes the signed distance of each point sample in 3D from the boundary of the underlying shape. Thus, given a set of signed distance values, the shape can be extracted by identifying the iso-surface using methods such as *Marching Cubes* [17]. As illustrated in Figure 4, given a convolutional neural network (CNN) that encodes the input image into a feature vector, DISN predicts the SDF value of a given 3D point using this feature vector. By sampling different 3D point locations, DISN is able to generate an implicit field of the underlying surface with infinite resolution. Moreover, without the need of a fixed topology assumption, the regressing target for DISN is an accurate ground truth instead of an approximated metric.

While many single-view 3D reconstruction methods [2, 10, 16, 1] that learn a shape embedding from a 2D image are able to capture the global shape properties, they have a tendency to ignore details such as holes or thin structures. Such fine-grained details only occupy a small portion in 3D space and thus sacrificing them does not incur a high loss compared to ground truth shape. However, such results can be visually unsatisfactory.

To address this problem, we introduce a local feature extraction module. Specifically, we estimate the viewpoint parameters of the input image. We utilize this information to project each query point onto the input image to identify a corresponding local patch. We extract local features from such patches and use them in conjunction with global image features to predict the SDF values of the 3D points. This module enables the network to learn the relations between projected pixels and 3D space and significantly improves the reconstruction quality of fine-grained details in the resulting 3D shape. As shown in Figure 1, DISN is able to generate shape details, such as the patterns on the bench back and holes on the rifle handle, which previous state-of-the-art methods fail to produce. To the best of our knowledge, DISN is the first deep learning model that is able to capture such high-quality details from single-view images.

We evaluate our approach on various shape categories using both synthetic data generated from 3D shape datasets as well as online product images. Qualitative and quantitative comparisons demonstrate that our network outperforms state-of-the-art methods and generates plausible shapes with high-quality details. Furthermore, we also extend DISN to multi-view reconstruction and other applications such as shape interpolation.

## 2   Related Work

There have been extensive studies on learning-based single-view 3D reconstruction using various 3D representations including voxels [2–8], octrees [18–20], points [10], and primitives [21, 22]. More recently, Sinha et al. [23] propose to generate the surface of an object using geometry images. Tang et al. [24] use shape skeletons for surface reconstruction, however, their method requires additional shape primitives dataset. Groueix et al. [11] present AtlasNet to generate surfaces of 3D shapes using a set of parametric surface elements. Wang et al. [12] introduce a graph-based network Pix2Mesh to reconstruct 3D manifold shapes from input images whereas Wang et al. [13] present 3DN to reconstruct a 3D shape by deforming a given source mesh.

Most of the aforementioned methods use explicit 3D representations and often suffer from problems such as limited resolution and fixed mesh topology. Implicit representations provide an alternative representation to overcome these limitations. In our work, we adopt the Signed Distance Functions (SDF) which are among the most popular implicit surface representations. Several deep learning approaches have utilized SDFs recently. Dai et al. [14] use a voxel-based SDF representation for shape inpainting. Nevertheless, 3D CNNs are known to suffer from high memory usage and computation cost. Park et al. [15] introduce DeepSDF for shape completion using an auto-decoder structure. However, their network is not feed-forward and requires optimizing the embedding vector during test time which limits the efficiency and capability of the approach. Chen and Zhang [16] use SDFs in deep networks for the task of shape generation. While their method achieves promising results for the

generation task, it fails to recover fine-grained details of 3D objects for single-view reconstruction. Mescheder et al. [1] learns an implicit representation by predicting the probability of each cell in a volumetric grid being occupied or not, i.e., being inside or outside of a 3D model. By iteratively subdividing each active cell (i.e., cells surrounded by occupied and empty cells) into sub-cells and repeating the prediction for each sub-cell, they alleviate the problem of the limited resolution of volumetric grids. Finally, in concurrent work, Saito et al. [25] utilize local image features to predict if a 3D point sample is inside or outside the surface of a mesh and demonstrate high quality human reconstruction results. In contrast, our method not only predicts the sign (i.e., being inside or outside) of sampled points but also the distance which is continuous. We compare our method with recent approaches in Section 4.1 and demonstrate state-of-the-art results.

## 3 Method

Given an image of an object, our goal is to reconstruct a 3D shape that captures both the overall structure and fine-grained details of the object. We consider modeling a 3D shape as a signed distance function (SDF). As illustrated in Figure 2, SDF is a continuous function that maps a given spatial point $\mathbf{p} = (x, y, z) \in \mathbb{R}^3$ to a real value $s \in \mathbb{R}$: $s = SDF(\mathbf{p})$. Instead of more common 3D representations such as depth [26], the absolute value of $s$ indicates the distance of the point to the surface, while the sign of $s$ represents if the point is inside or outside the surface. An iso-surface $\mathcal{S}_0 = \{\mathbf{p} | SDF(\mathbf{p}) = 0\}$ implicitly represents the underlying 3D shape.

In this paper, we use a feed-forward deep neural network, Deep Implicit Surface Network (DISN), to predict the SDF from an input image. DISN takes a single image as input and predicts the SDF value for any given point. Unlike the 3D CNN methods [14] which generate a volumetric grid with fixed resolution, DISN produces a continuous field with arbitrary resolution. Moreover, we introduce a local feature extraction method to improve recovery of shape details.

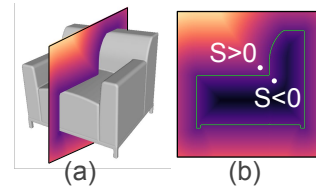

(a)          (b)

Figure 2: Illustration of SDF. (a) Rendered 3D surface with $s = 0$. (b) Cross-section of the SDF. A point is outside the surface if $s > 0$, inside if $s < 0$, and on the surface if $s = 0$.

### 3.1 DISN: Deep Implicit Surface Network

The overview of our method is illustrated in Figure 4. Given an image, DISN consists of two parts: *camera pose estimation* and *SDF prediction*. DISN first estimates the camera parameters that map an object in world coordinates to the image plane. Given the predicted camera parameters, we project each 3D query point onto the image plane and collect multi-scale CNN features for the corresponding image patch. DISN then decodes the given spatial point to an SDF value using both the multi-scale local image features and the global image features.

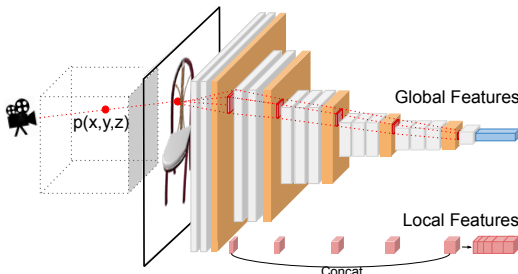

Figure 3: Local feature extraction. Given a 3D point $\mathbf{p}$, we use the estimated camera parameters to project $\mathbf{p}$ onto the image plane. Then we identify the projected location on each feature map layer of the encoder. We concatenate features at each layer to get the local features of point $\mathbf{p}$.

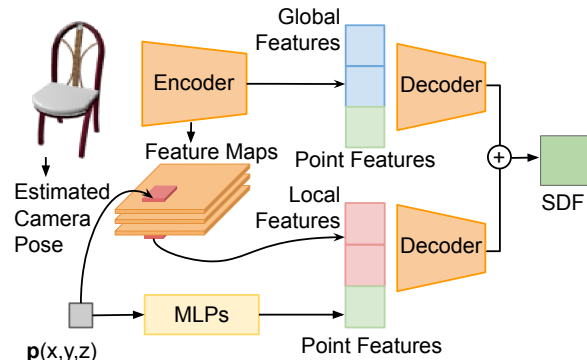

Figure 4: Given an image and a point $\mathbf{p}$, we estimate the camera pose and project $\mathbf{p}$ onto the image plane. DISN uses the local features at the projected location, the global features, and the point features to predict the SDF of $\mathbf{p}$. 'MLPs' denotes multi-layer perceptrons.

#### 3.1.1 Camera Pose Estimation

Given an input image, our first goal is to estimate the corresponding viewpoint. We train our network on the ShapeNet Core dataset [27] where all the models are aligned. Therefore we use this aligned

model space as the world space where our camera parameters are with respect to, and we assume a fixed set of intrinsic parameters. Regressing camera parameters from an input image directly using a CNN often fails to produce accurate poses as discussed in [28]. To overcome this issue, Insafutdinov and Dosovitskiy [28] introduce a distilled ensemble approach to regress camera pose by combining several pose candidates. However, this method requires a large number of network parameters and a complex training procedure. We present a more efficient and effective network illustrated in Figure 5. In a recent work, Zhou et al. [29] show that a $6D$ rotation representation is continuous and easier for a neural network to regress compared to more commonly used representations such as quaternions and Euler angles. Thus, we employ the $6D$ rotation representation $\mathbf{b} = (\mathbf{b_x}, \mathbf{b_y})$, where $\mathbf{b} \in \mathbb{R}^6, \mathbf{b_x} \in \mathbb{R}^3, \mathbf{b_y} \in \mathbb{R}^3$. Given $\mathbf{b}$, the rotation matrix $\mathbf{R} = (\mathbf{R_x}, \mathbf{R_y}, \mathbf{R_z})^T \in \mathbb{R}^{3 \times 3}$ is obtained by

$$\mathbf{R_x} = N(\mathbf{b_x}), \mathbf{R_z} = N(\mathbf{R_x} \times \mathbf{b_y}), \mathbf{R_y} = \mathbf{R_z} \times \mathbf{R_x}, \tag{1}$$

where $\mathbf{R_x}, \mathbf{R_y}, \mathbf{R_z} \in \mathbb{R}^3$, $N(\cdot)$ is the normalization function, '$\times$' indicates cross product. Translation $\mathbf{t} \in \mathbb{R}^3$ from world space to camera space is directly predicted by the network.

Instead of calculating losses on camera parameters directly as in [28], we use the predicted camera pose to transform a given point cloud from the world space to the camera coordinate space. We compute the loss $L_{cam}$ by calculating the mean squared error between the transformed point cloud and the ground truth point cloud in the camera space:

$$L_{cam} = \frac{\sum_{\mathbf{p}_w \in PC_w} ||\mathbf{p}_G - (\mathbf{R}\mathbf{p}_w + \mathbf{t}))||_2^2}{\sum_{\mathbf{p}_w \in PC_w} 1}, \tag{2}$$

where $PC_w \in \mathbb{R}^{N \times 3}$ is the point cloud in the world space, $N$ is number of points in $PC_w$. For each $\mathbf{p}_w \in PC_w$, $\mathbf{p}_G$ represents the corresponding ground truth point location in the camera space and $|| \cdot ||_2^2$ is the squared $L_2$ distance.

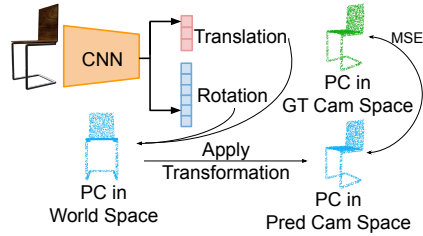

Figure 5: Camera Pose Estimation Network. 'PC' denotes point cloud. 'GT Cam' and 'Pred Cam' denote the ground truth and predicted cameras.

### 3.1.2 SDF Prediction with Deep Neural Network

Given an image $I$, we denote the ground truth SDF by $SDF^I(\cdot)$, and the goal of our network $f(\cdot)$ is to estimate $SDF^I(\cdot)$. Unlike the commonly used CD and EMD losses in previous reconstruction methods [10, 11], our guidance is a true ground truth instead of approximated metrics.

Park et al [15] recently propose DeepSDF, a direct approach to regress SDF with a neural network. DeepSDF concatenates the location of a query 3D point and the shape embedding extracted from a depth image or a point cloud and uses an auto-decoder to obtain the corresponding SDF value. The auto-decoder structure requires optimizing the shape embedding for each object. In our initial experiments, when we applied a similar network architecture in a feed-forward manner, we observed convergence issues. Alternatively, Chen and Zhang [16] propose to concatenate the global features of an input image and the location of a query point to every layer of a decoder. While this approach works better in practice, it also results in a significant increase in the number of network parameters. Our solution is to use a multi-layer perceptron to map the given point location to a higher-dimensional feature space. This high dimensional feature is then concatenated with global and local image features respectively and used to regress the SDF value. We provide the details of our network in the supplementary.

**Local Feature Extraction** As shown in Figure 6(a), our initial experiments showed that it is hard to capture shape details such as holes and thin structures when only global image features are used. Thus, we introduce a local feature extraction method to focus on reconstructing fine-grained details, such as the back poles of a chair (Figure 6). As illustrated in Figure 3, a 3D point $\mathbf{p} \in \mathbb{R}^3$ is projected to a 2D location $\mathbf{q} \in \mathbb{R}^2$ on the image plane with the estimated camera parameters. We retrieve features on each feature map corresponding to location $\mathbf{q}$ and concatenate them to get the local image features. Since the feature maps in the later layers are smaller in dimension than the original image, we resize them to the original size with bilinear interpolation and extract the resized features at location $\mathbf{q}$.

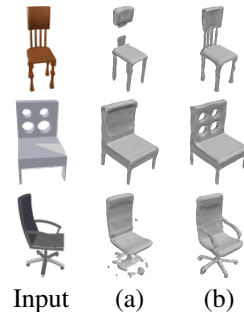

Input (a) (b)

Figure 6: Shape reconstruction results (a) without and (b) with local feature extraction.

Two decoders then take the global and local image features respectively as input with the point features and make an SDF prediction. The final SDF is the sum of these two predictions. Figure 6 compares the results of our approach with and without local feature extraction. With only global features, the network is able to predict the overall shape but fails to produce details. Local feature extraction helps to recover these missing details by predicting the *residual* SDF.

**Loss Functions**   We regress continuous SDF values instead of formulating a binary classification problem (e.g., inside or outside of a shape) as in [16]. This strategy enables us to extract surfaces that correspond to different iso-values. To ensure that the network concentrates on recovering the details near and inside the iso-surface $\mathcal{S}_0$, we propose a weighted loss function. Our loss is defined by

$$L_{SDF} = \sum_{\mathbf{p}} m|f(I, \mathbf{p}) - SDF^I(\mathbf{p})|,$$

$$m = \begin{cases} m_1, & \text{if } SDF^I(\mathbf{p}) < \delta, \\ m_2, & \text{otherwise,} \end{cases} \tag{3}$$

where $|\cdot|$ is the $L_1$-norm. $m_1$, $m_2$ are different weights, and for points whose signed distance is below a certain threshold $\delta$, we use a higher weight of $m_1$.

### 3.2   Surface Reconstruction

To generate a mesh surface, we firstly define a dense 3D grid and predict SDF values for each grid point. Once we compute the SDF values for each point in the dense grid, we use Marching Cubes [17] to obtain the 3D mesh that corresponds to the iso-surface $\mathcal{S}_0$.

## 4   Experiments

We perform quantitative and qualitative comparisons on single-view 3D reconstruction with state-of-the-art methods [11–13, 16, 1] in Section 4.1. We also compare the performance of our method on

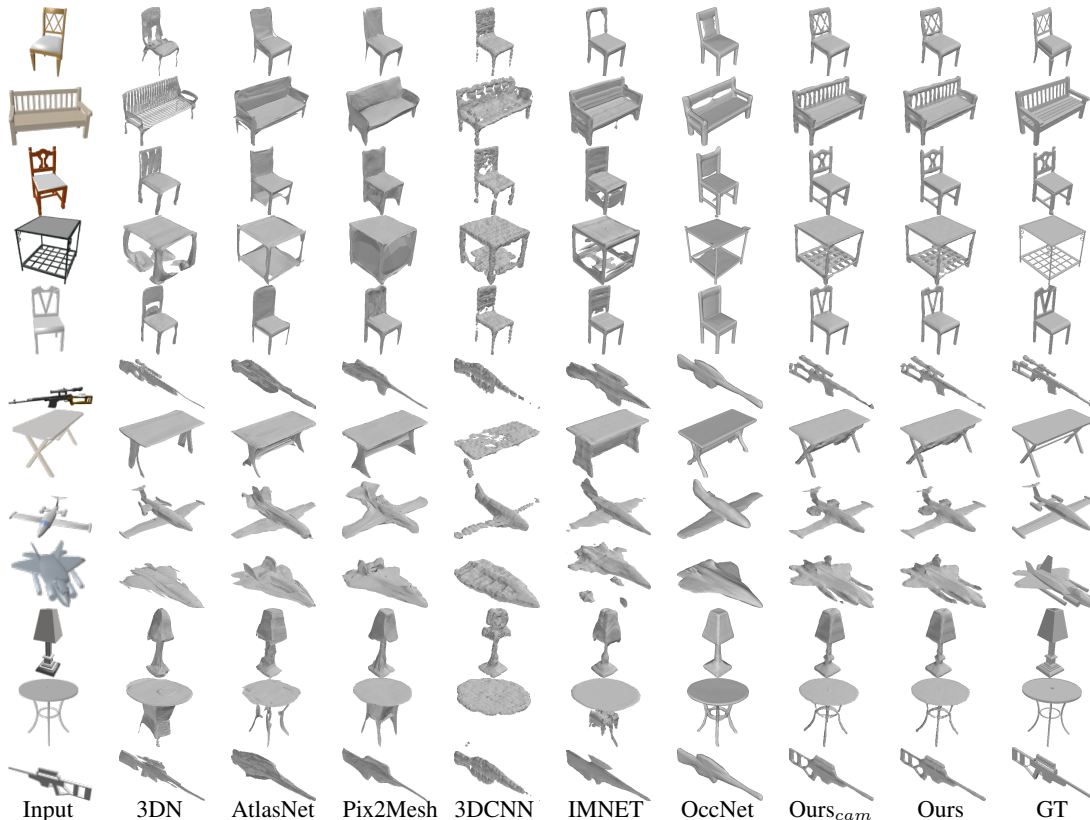

Input      3DN      AtlasNet      Pix2Mesh      3DCNN      IMNET      OccNet      Ours$_{cam}$      Ours      GT

Figure 7: Single-view reconstruction results of various methods. 'GT' denotes ground truth shapes. Best viewed on screen with zooming in.

camera pose estimation with [28] in Section 4.2. We further conduct ablation studies in Section 4.3 and showcase several applications in Section 4.4. More qualitative results and all detailed network architectures can be found in supplementary.

**Dataset**   For both camera prediction and SDF prediction, we follow the settings of [11–13, 1], and use the ShapeNet Core dataset [27], which includes 13 object categories, and an official training/testing split to train and test our method. We train a single network on all categories and report the test results generated by this network.

Choy et al. [30] provide a dataset of renderings of ShapeNet Core models where each model is rendered from 24 views with limited variation in terms of camera orientation. In order to make our method more general, we provide a new 2D dataset [1] composed of renderings of the models in ShapeNet Core. Specifically, for each mesh model, our dataset provides 36 renderings with smaller variation(similar to [30]'s) and 36 views with a larger variation(bigger yaw angle range and larger distance variation). Unlike Choy et al., we allow the object to move away from the origin, therefore, providing more degrees of freedom in terms of camera parameters. We ignore the "Roll" angle of the camera since it is very rare in a real-world scenarios. We also render higher resolution images (224 by 224 instead of the original 137 by 137). Finally, to facilitate future studies, we also pair each rendered RGBA image with a depth image, a normal map and an albedo image as shown in Figure 8.

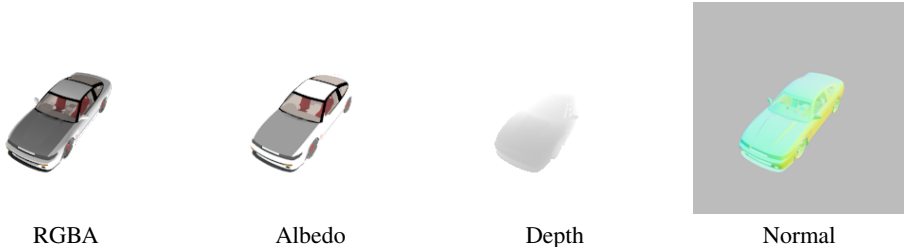

| RGBA | Albedo | Depth | Normal |

Figure 8: Each view of each object has four representations correspondingly

**Data Preparation and Implementation Details**   For each 3D mesh in ShapeNet Core, we first generate an SDF grid with resolution $256^3$ using [31, 32]. Models in ShapeNet Core are aligned and we choose this aligned model space as our world space where each render view in [30] represents a transformation to a different camera space.

We train our camera pose estimation network and SDF prediction network separately. For both networks, we use VGG-16 [33] as the image encoder. When training the SDF prediction network, we extract the local features using the ground truth camera parameters. As mentioned in Section 3.1, DISN is able to generate a signed distance field with an arbitrary resolution by continuously sampling points and regressing their SDF values. However, in practice, we are interested in points near the iso-surface $\mathcal{S}_0$. Therefore, we use Monte Carlo sampling to choose 2048 grid points under Gaussian distribution $\mathcal{N}(0, 0.1)$ during training. We choose $m_1 = 4$, $m_2 = 1$, and $\delta = 0.01$ as the parameters of Equation 3. Our network is implemented with TensorFlow. We use the Adam optimizer with a learning rate of $1 \times 10^{-4}$ and a batch size of 16.

For testing, we first use the camera pose prediction network to estimate the camera parameters for the input image and feed the estimated parameters as input to SDF prediction. We follow the aforementioned surface reconstruction procedure (Section 3.2) to generate the output mesh.

**Evaluation Metrics**   For quantitative evaluations, we apply four commonly used metrics to compute the difference between a reconstructed mesh object and its ground truth mesh: (1) Chamfer Distance (CD), (2) Earth Mover's Distance (EMD) between uniformly sampled point clouds, (3) Intersection over Union (IoU) on voxelized meshes, and (4) F-Score [34]. The definitions of CD and EMD can be found in the supplemental.

## 4.1   Single-view Reconstruction Comparison With State-of-the-art Methods

In this section, we compare our approach on single-view reconstruction with state-of-the-art methods: AtlasNet [11], Pixel2Mesh [12], 3DN [13], OccNet [1] and IMNET [16]. AtlasNet [11] and

Pixel2Mesh [12] generate a fixed-topology mesh from a 2D image. 3DN [13] deforms a given source mesh to reconstruct the target model. When comparing to this method, we choose a source mesh from a given set of templates by querying a template embedding as proposed in the original work. IMNET [16] and OccNet [1] both predict the sign of SDF to reconstruct 3D shapes. Since IMNET trains an individual model for each category, we implement their model following the original paper and train a single model on all 13 categories. Due to mismatch between the scales of shapes reconstructed by our method and OccNet, we only report their IoU, which is scale-invariant. In addition, we train a 3D CNN model, denoted by '3DCNN', where the encoder is the same as DISN and a decoder is a volumetric 3D CNN structure with an output dimension of $64^3$. The ground truth for 3DCNN is the SDF values on all $64^3$ grid locations. For both IMNET and 3DCNN, we use the same surface reconstruction method as ours to output reconstructed meshes. We also report the results of DISN using estimated camera poses and ground truth poses, denoted by 'Ours$_{cam}$' and 'Ours' respectively. AtlasNet, Pixel2Mesh, and 3DN use explicit surface generation, while 3DCNN, IMNET, OccNet, and our methods reconstruct implicit surfaces.

As shown in Table 1, DISN outperforms all other models in EMD and IoU. Only 3DN performs better than our model on CD, however, 3DN requires more information than ours in the form of a source mesh as input. Figure 7 shows qualitative results. As illustrated in both quantitative and qualitative results, implicit surface representation provides a flexible method of generating topology-variant 3D meshes. Comparisons to 3D CNN show that predicting SDF values for given points produces smoother surfaces than generating a fixed 3D volume using an image embedding. We speculate that this is due to SDF being a continuous function with respect to point locations. It is harder for a deep network to approximate an overall SDF volume with global image features only. Moreover, our method outperforms IMNET and OccNet in terms of recovering shape details. For example, in Figure 7, local feature extraction enables our method to generate different patterns of the chair backs in the first three rows, while other methods fail to capture such details. We further validate the effectiveness of our local feature extraction module in Section 4.3. Although using ground truth camera poses (i.e., 'Ours') outperforms using predicted camera poses (i.e., 'Ours$_{cam}$') in quantitative results, respective qualitative results demonstrate no significant difference.

| | | plane | bench | box | car | chair | display | lamp | speaker | rifle | sofa | table | phone | boat | Mean |
|---|---|---|---|---|---|---|---|---|---|---|---|---|---|---|---|
| EMD | AtlasNet | 3.39 | 3.22 | 3.36 | 3.72 | 3.86 | 3.12 | 5.29 | 3.75 | 3.35 | 3.14 | 3.98 | 3.19 | 4.39 | 3.67 |
| | Pxl2mesh | 2.98 | 2.58 | 3.44 | 3.43 | 3.52 | 2.92 | 5.15 | 3.56 | 3.04 | 2.70 | 3.52 | 2.66 | 3.94 | 3.34 |
| | 3DN | 3.30 | 2.98 | 3.21 | 3.28 | 4.45 | 3.91 | **3.99** | 4.47 | 2.78 | 3.31 | 3.94 | 2.70 | 3.92 | 3.56 |
| | IMNET | 2.90 | 2.80 | 3.14 | 2.73 | 3.01 | 2.81 | 5.85 | 3.80 | 2.65 | 2.71 | 3.39 | 2.14 | 2.75 | 3.13 |
| | 3D CNN | 3.36 | 2.90 | 3.06 | **2.52** | 3.01 | 2.85 | 4.73 | **3.35** | 2.71 | **2.60** | **3.09** | 2.10 | **2.67** | 3.00 |
| | Ours$_{cam}$ | **2.67** | **2.48** | **3.04** | 2.67 | **2.67** | **2.73** | 4.38 | 3.47 | **2.30** | 2.62 | 3.11 | **2.06** | 2.77 | **2.84** |
| | Ours | 2.45 | 2.41 | 2.99 | 2.52 | 2.62 | 2.63 | 4.11 | 3.37 | 1.93 | 2.55 | 3.07 | 2.00 | 2.55 | 2.71 |
| CD | AtlasNet | **5.98** | 6.98 | 13.76 | 17.04 | 13.21 | 7.18 | 38.21 | 15.96 | 4.59 | 8.29 | 18.08 | 6.35 | 15.85 | 13.19 |
| | Pxl2mesh | 6.10 | **6.20** | 12.11 | 13.45 | 11.13 | **6.39** | 31.41 | **14.52** | 4.51 | **6.54** | 15.61 | 6.04 | 12.66 | 11.28 |
| | 3DN | 6.75 | 7.96 | **8.34** | 7.09 | 17.53 | 8.35 | **12.79** | 17.28 | **3.26** | 8.27 | 14.05 | **5.18** | 10.20 | **9.77** |
| | IMNET | 12.65 | 15.10 | 11.39 | 8.86 | 11.27 | 13.77 | 63.84 | 21.83 | 8.73 | 10.30 | 17.82 | 7.06 | 13.25 | 16.61 |
| | 3D CNN | 10.47 | 10.94 | 10.40 | **5.26** | 11.15 | 11.78 | 35.97 | 17.97 | 6.80 | 9.76 | **13.35** | 6.30 | **9.80** | 12.30 |
| | Ours$_{cam}$ | 9.96 | 8.98 | 10.19 | 5.39 | **7.71** | 10.23 | 25.76 | 17.90 | 5.58 | 9.16 | 13.59 | 6.40 | 11.91 | 10.98 |
| | Ours | 9.01 | 8.32 | 9.98 | 4.92 | 7.54 | 9.58 | 22.73 | 16.70 | 4.36 | 8.71 | 13.29 | 6.21 | 10.87 | 10.17 |
| IoU | AtlasNet | 39.2 | 34.2 | 20.7 | 22.0 | 25.7 | 36.4 | 21.3 | 23.2 | 45.3 | 27.9 | 23.3 | 42.5 | 28.1 | 30.0 |
| | Pxl2mesh | 51.5 | 40.7 | 43.4 | 50.1 | 40.2 | 55.9 | 29.1 | 52.3 | 50.9 | 60.0 | 31.2 | 69.4 | 40.1 | 47.3 |
| | 3DN | 54.3 | 39.8 | 49.4 | 59.4 | 34.4 | 47.2 | 35.4 | 45.3 | 57.6 | 60.7 | 31.3 | 71.4 | 46.4 | 48.7 |
| | IMNET | 55.4 | 49.5 | 51.5 | 74.5 | 52.2 | 56.2 | 29.6 | 52.6 | 52.3 | 64.1 | 45.0 | 70.9 | 56.6 | 54.6 |
| | 3D CNN | 50.6 | 44.3 | 52.3 | **76.9** | 52.6 | 51.5 | 36.2 | 58.0 | 50.5 | **67.2** | 50.3 | 70.9 | **57.4** | 55.3 |
| | OccNet | 54.7 | 45.2 | **73.2** | 73.1 | 50.2 | 47.9 | **37.0** | **65.3** | 45.8 | 67.1 | **50.6** | 70.9 | 52.1 | 56.4 |
| | **DISN**$_{cam}$ | **57.5** | **52.9** | 52.3 | 74.3 | **54.3** | **56.4** | 34.7 | 54.9 | **59.2** | 65.9 | 47.9 | **72.9** | 55.9 | **57.0** |
| | **DISN** | 61.7 | 54.2 | 53.1 | 77.0 | 54.9 | 57.7 | 39.7 | 55.9 | 68.0 | 67.1 | 48.9 | 73.6 | 60.2 | 59.4 |

Table 1: Quantitative results on ShapeNet Core for various methods. Metrics are CD ($\times 0.001$, the smaller the better), EMD ($\times 100$, the smaller the better) and IoU (%, the larger the better). CD and EMD are computed on 2048 points.

We also compute the F-score (see Table 2) which measures the percentage of surface area that is reconstructed correctly and thus provides a reliable metric [34]. In our evaluations, we use

| Threshold(%) | 0.5% | 1% | 2% | 5% | 10% | 20% |
|---|---|---|---|---|---|---|
| 3DCNN | 0.064 | 0.295 | 0.691 | 0.935 | 0.984 | 0.997 |
| IMNet | 0.063 | 0.286 | 0.673 | 0.922 | 0.977 | 0.995 |
| DISN | 0.079 | 0.327 | 0.718 | 0.943 | 0.984 | 0.996 |
| DISN$_{cam}$ | 0.070 | 0.307 | 0.700 | 0.940 | 0.986 | 0.998 |

Table 2: F-Score for varying thresholds (% of reconstruction volume side length, same as [34]) on all categories.

|  | [28] | Ours | Ours$_{new}$ |
|---|---|---|---|
| $d_{3D}$ | 0.073 | **0.047** | 0.059 |
| $d_{2D}$ | 4.86 | **2.95** | 4.38/2.67 |

Table 3: Camera pose estimation comparison. The unit of $d_{2D}$ is pixels.

$F_1 = 2 * (\text{Precision} \cdot \text{Recall})/(\text{Precision} + \text{Recall})$. We uniformly sample points from both ground truth and generated meshes. We define precision as the percentage of the generated points whose distance to the closest ground truth point is less than a threshold. Similarly, we define recall as the percentage of ground truth points whose distance to the closest generated point is less than a threshold.

## 4.2 Camera Pose Estimation

We compare our camera pose estimation with [28]. Given a point cloud $PC_w$ in world coordinates for an input image, we transform $PC_w$ using the predicted camera pose and compute the mean distance $d_{3D}$ between the transformed point cloud and the ground truth point cloud in camera space. We also compute the 2D reprojection error $d_{2D}$ of the transformed point cloud after we project it onto the input image. Table 3 reports $d_{3D}$ and $d_{2D}$ of [28] and our method. With the help of the $6D$ rotation representation, our method outperforms [28] by 2 pixels in terms of 2D reprojection error. We also train and test the pose estimation on the new 2D dataset. Even these images possess more view variation, because of the better rendering quality, we can achieve an average 2D distance of 4.38 pixels on 224 by 224 images (2.67 pixels if normalized to the original resolution of 137 by 137).

## 4.3 Ablation Studies

To show the impact of the camera pose estimation, local feature extraction, and different network architectures, we conduct ablation studies on the ShapeNet "chair" category, since it has the greatest variety. Table 4 reports the quantitative results and Figure 9 shows the qualitative results.

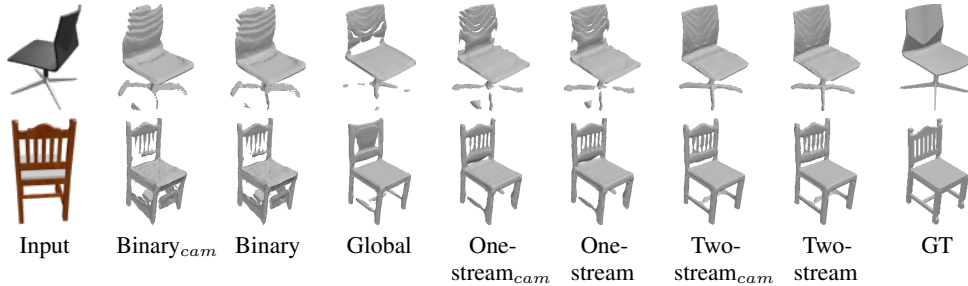

|       Input       Binary$_{cam}$   Binary   Global   One-              One-       Two-             Two-       GT |
|---|
|                                                              stream$_{cam}$    stream    stream$_{cam}$    stream |

Figure 9: Qualitative results of our method using different settings. 'GT' denotes ground truth shapes, and '$_{cam}$' denotes models with estimated camera parameters.

**Camera Pose Estimation** As is shown in Section 4.2, camera pose estimation potentially introduces uncertainty to the local feature extraction process with an average reprojection error of 2.95 pixels. Although the quantitative reconstruction results with ground truth camera parameters are constantly superior to the results with estimated parameters in Table 4, Figure 9 demonstrates that a small difference in the image projection does not affect the reconstruction quality significantly.

**Binary Classification** Previous studies [1, 16] formulate SDF prediction as a binary classification problem by predicting the probability of a point being inside or outside the surface $\mathcal{S}_0$. Even though Section 4.1 illustrates our superior performance over [1, 16], we further validate the effectiveness of our regression supervision by comparing with classification supervision using our own network structure. Instead of producing a SDF value, we train our network with classification supervision and output the probability of a point being inside the mesh surface. We use a softmax cross entropy loss to optimize this network. We report the result of this classification network as 'Binary'.

**Local Feature Extraction** Local image features of each point provide access to the corresponding local information that captures shape details. To validate the effectiveness of this information, we remove the 'local features extraction' module from DISN and denote this setting by 'Global'. This

model predicts the SDF value solely based on the global image features. By comparing 'Global' with other methods in Table 4 and Figure 9, we conclude that local feature extraction helps the model capture shape details and improve the reconstruction quality by a large margin.

**Network Structures**  To further assess the impact of different network architectures, in addition to our original architecture with two decoders (which we call 'Two-stream'), we also introduce a 'One-stream' architecture where the global features, the local features, and the point features are concatenated and fed into a single decoder which predicts the SDF value. Detailed structure of this architecture can be found in the supplementary. As illustrated in Table 4 and Figure 9, the original Two-stream setting is slightly superior to One-stream, which shows that DISN is robust to different network architectures.

| Camera Pose | Binary ground truth \| estimated | Global n/a | One-stream ground truth \| estimated | Two-stream ground truth \| estimated |
|---|---|---|---|---|
| EMD | 2.88 \| 2.99 | 2.75 \| n/a | 2.71 \| 2.74 | **2.62 \| 2.65** |
| CD | 8.27 \| 8.80 | 7.64 \| n/a | 7.86 \| 8.30 | **7.55 \| 7.63** |
| IoU | 54.9 \| 53.5 | 54.8 \| n/a | 53.6 \| 53.5 | **55.3 \| 53.9** |

Table 4: Quantitative results on the category "chair". CD ($\times 0.001$), EMD ($\times 100$) and IoU (%).

## 4.4 Applications

**Shape interpolation**  Figure 10 shows shape interpolation results where we interpolate both global and local image features going from the leftmost sample to the rightmost. We see that the generated shape is gradually transformed.

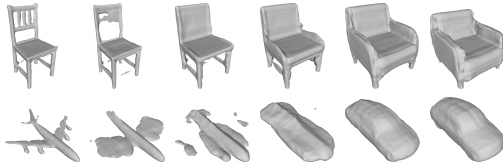

Figure 10: Shape interpolation result.

**Test with online product images**  Figure 11 illustrates 3D reconstruction results by DISN on online product images. Note that our model is trained on rendered images, this experiment validates the domain transferability of DISN.

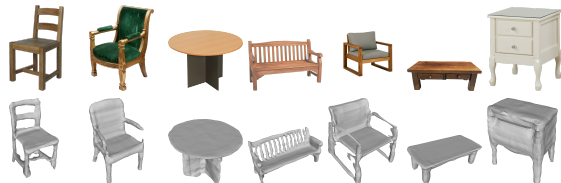

Figure 11: Test our model on online product images.

**Multi-view reconstruction**  Our model can also take multiple 2D views of the same object as input. After extracting the global and the local image features for each view, we apply max pooling and use the resulting features as input to each decoder. We have retrained our network for 3 input views and visualize some results in Figure 12. Combining multi-view features helps DISN to further address shape details.

## 5 Conclusion

In this paper, we present DISN, a deep implicit surface network for single-view reconstruction. Given a 3D point and an input image, DISN predicts the SDF value for the point. We introduce a local feature extraction module by projecting the 3D point onto the image plane with an estimated camera pose. With the help of such local features, DISN is able to capture fine-grained details and generate high-quality 3D models. Qualitative and quantitative experiments validate the superior performance of DISN over state-of-the-art methods and the flexibility of our model.

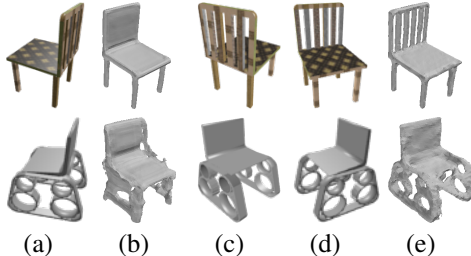

| (a) | (b) | (c) | (d) | (e) |

Figure 12: Multi-view reconstruction results. (a) Single-view input. (b) Reconstruction result from (a). (c)&(d) Two other views. (e) Multi-view reconstruction result from (a), (c) and (d).

Though we achieve state-of-the-art performance in single-view reconstruction, our method is only able to handle objects with a clear background since it's trained with rendered images. To address this limitation, our future work includes extending SDF generation with texture prediction using a differentiable renderer [35].

## Footnotes

* indicates equal contributions.

[1]\texttt{https://github.com/Xharlie/ShapenetRender\_more\_variation}

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
