[Supplementary Material]

# Supplementary for DISN

## 1 Network Structures

The encoders are all VGG-16. To get the local features in corresponding location, we reshape the feature maps to the the original image size with biliear interpolation.

### 1.1 DISN Camera Prediction

The detail network architecture for camera prediction is illustrated in Figure I.

### 1.2 DISN Two-stream

The detail network architecture for the two-stream network is illustrated in Figure II.

### 1.3 DISN One-stream

The detail network architecture for the one-stream network is illustrated in Figure III.

### 1.4 Multi-view Architecture

We use the one-stream network as is shown in Figure III to compute the embedding vector for every view. A max-pooling operation is followed by the embedding vectors, and we use the decoder as in Figure III to compute the predicted SDF value.

## 2 Evaluation Metrics

We use Chamfer distance (CD) and Earth mover's distance (EMD) as our evaluation metrics. Suppose $PC$ and $PC_T$ are the sampled point clouds from predicted mesh and ground truth mesh respectively, CD is defined by

$$
\begin{aligned}
CD(PC, PC_T) = &\sum_{p_1 \in PC} \min_{p_2 \in PC_T} \|p_1 - p_2\|_2^2 \\
&+ \sum_{p_2 \in PC_T} \min_{p_1 \in PC} \|p_1 - p_2\|_2^2,
\end{aligned}
\tag{1}
$$

and EMD is defined by

$$
EMD(PC, PC_T) = \min_{\phi: PC \to PC_T} \sum_{p \in PC} \|p - \phi(p)\|_2,
\tag{2}
$$

where $\phi : PC \to PC_T$ is a bijection (one-to-one matching).

Figure I: Camera Pose Prediction Network.

Figure II: DISN Two-stream Network.

Figure III: DISN One-stream Network.

Input     3DN     AtlasNet     Pix2Mesh     3DCNN     IMNET     Ours     $\text{Ours}_{cam}$     GT

Figure IV: More qualitative results.

## 3    Computation Efficiency

It is possible to predict SDF values only for a small set of uniformly sampled points and apply trilinear interpolation to recover a dense grid of signed distance values. However, we observe that with the computation power of GPUs, predicting SDF at densely sampled points does not incur additional computation cost.

Suppose the target model can be represented by a voxelized grid of size $N \times N \times N$, the complexity of our algorithm is $O(N^3)$ since we predict the SDF value for each grid cell. We test our model on an Intel i7 machine with Geforce 1080Ti GPU. Network convergence takes 50 epochs. The average inference time (SDF generation and Marching Cube) for a single mesh of DISN Two-stream and OccNet is reported in Table I. DISN is much faster than the OccNet in terms of computing speed.

## 4    More qualitative results

Figure IV to XXXIV illustrate more qualitative results for single-view reconstruction of DISN and other methods. 'GT' denotes ground truth 3D mesh. Input images are ShapeNet rendered images. As is shown in these figures, our methods constantly outperform state-of-the-art methods.

| Resolution | OccNet | DISN |
|:---:|:---:|:---:|
| $64^3$ | 6.00 | 0.27 |
| $256^3$ | 148 | 27 |

Table I: Average inference time (seconds) for a single mesh.

| Input | 3DN | AtlasNet | Pix2Mesh | 3DCNN | IMNET | Ours | Ours$_{cam}$ | GT |

Figure V: More qualitative results.

| Input | 3DN | AtlasNet | Pix2Mesh | 3DCNN | IMNET | Ours | Ours$_{cam}$ | GT |

Figure VI: More qualitative results.

Figure VII: More qualitative results.

Input    3DN    AtlasNet    Pix2Mesh    3DCNN    IMNET    Ours    Ours$_{cam}$    GT

| Input | 3DN | AtlasNet | Pix2Mesh | 3DCNN | IMNET | Ours | Ours$_{cam}$ | GT |

Figure VIII: More qualitative results.

| Input | 3DN | AtlasNet | Pix2Mesh | 3DCNN | IMNET | Ours | Ours$_{cam}$ | GT |

Figure IX: More qualitative results.

| Input | 3DN | AtlasNet | Pix2Mesh | 3DCNN | IMNET | Ours | Ours$_{cam}$ | GT |

Figure X: More qualitative results.

| Input | 3DN | AtlasNet | Pix2Mesh | 3DCNN | IMNET | Ours | Ours$_{cam}$ | GT |

Figure XI: More qualitative results.

Input     3DN     AtlasNet     Pix2Mesh     3DCNN     IMNET     Ours     Ours$_{cam}$     GT

Figure XII: More qualitative results.

| Input | 3DN | AtlasNet | Pix2Mesh | 3DCNN | IMNET | Ours | Ours$_{cam}$ | GT |

Figure XIII: More qualitative results.

Input     3DN     AtlasNet     Pix2Mesh     3DCNN     IMNET     Ours     Ours$_{cam}$     GT

Figure XIV: More qualitative results.

Input    3DN    AtlasNet    Pix2Mesh    3DCNN    IMNET    Ours    Ours$_{cam}$    GT

Figure XV: More qualitative results.

| Input | 3DN | AtlasNet | Pix2Mesh | 3DCNN | IMNET | Ours | Ours$_{cam}$ | GT |

Figure XVI: More qualitative results.

| Input | 3DN | AtlasNet | Pix2Mesh | 3DCNN | IMNET | Ours | Ours$_{cam}$ | GT |

Figure XVII: More qualitative results.

| Input | 3DN | AtlasNet | Pix2Mesh | 3DCNN | IMNET | Ours | Ours$_{cam}$ | GT |

Figure XVIII: More qualitative results.

| Input | 3DN | AtlasNet | Pix2Mesh | 3DCNN | IMNET | Ours | Ours$_{cam}$ | GT |

Figure XIX: More qualitative results.

|  Input | 3DN | AtlasNet | Pix2Mesh | 3DCNN | IMNET | Ours | Ours$_{cam}$ | GT |

Figure XX: More qualitative results.

Input      3DN      AtlasNet      Pix2Mesh      3DCNN      IMNET      Ours      Ours$_{cam}$      GT

Figure XXI: More qualitative results.

| Input | 3DN | AtlasNet | Pix2Mesh | 3DCNN | IMNET | Ours | Ours$_{cam}$ | GT |

Figure XXII: More qualitative results.

| Input | 3DN | AtlasNet | Pix2Mesh | 3DCNN | IMNET | Ours | Ours$_{cam}$ | GT |

Figure XXIII: More qualitative results.

| Input | 3DN | AtlasNet | Pix2Mesh | 3DCNN | IMNET | Ours | Ours$_{cam}$ | GT |
|-------|-----|----------|----------|-------|-------|------|----------|----|

Figure XXIV: More qualitative results.

Input    3DN    AtlasNet    Pix2Mesh    3DCNN    IMNET    Ours    Ours$_{cam}$    GT

Figure XXV: More qualitative results.

| Input | 3DN | AtlasNet | Pix2Mesh | 3DCNN | IMNET | Ours | Ours$_{cam}$ | GT |

Figure XXVI: More qualitative results.

Input    3DN    AtlasNet    Pix2Mesh    3DCNN    IMNET    Ours    Ours$_{cam}$    GT

Figure XXVII: More qualitative results.

| Input | 3DN | AtlasNet | Pix2Mesh | 3DCNN | IMNET | Ours | Ours$_{cam}$ | GT |

Figure XXVIII: More qualitative results.

| Input | 3DN | AtlasNet | Pix2Mesh | 3DCNN | IMNET | Ours | Ours$_{cam}$ | GT |

Figure XXIX: More qualitative results.

| Input | 3DN | AtlasNet | Pix2Mesh | 3DCNN | IMNET | Ours | Ours$_{cam}$ | GT |

Figure XXX: More qualitative results.

| Input | 3DN | AtlasNet | Pix2Mesh | 3DCNN | IMNET | Ours | Ours$_{cam}$ | GT |

Figure XXXI: More qualitative results.

Input     3DN     AtlasNet     Pix2Mesh     3DCNN     IMNET     Ours     $\text{Ours}_{cam}$     GT

Figure XXXII: More qualitative results.

| Input | 3DN | AtlasNet | Pix2Mesh | 3DCNN | IMNET | Ours | Ours$_{cam}$ | GT |

Figure XXXIII: More qualitative results.

| Input | 3DN | AtlasNet | Pix2Mesh | 3DCNN | IMNET | Ours | Ours$_{cam}$ | GT |

Figure XXXIV: More qualitative results.