[Reviews · NeurIPS 2019]

Reviewer 1



Update: The F-score results and the naive baseline shown in the rebuttal are good. I agree with R3 that an experiment with more camera variations should be added, which is also promised in the rebuttal. --- The paper presents a new method for single image 3D reconstruction. The method generates a signed distance function, which contains the surface as the zero level set. Implicit representations have very recently become popular for this task but the presented network architecture makes good use of this representation. The method works similarly to Occupancy networks, IMNET and DeepSDF. Given an image of an object, the method maps a 3D position to a signed distance value. To improve the reconstruction of details, this paper uses the camera pose (estimated by another network) to project each 3D position to the image plane and look up local image features generated by a VGG encoder. Making use of this local feature look-up distinguishes this work. The decoding stage then combines two SDF values generated by decoders processing the local and global features separately. Originality: The approach estimates the camera pose(i) and combines an implicit representation(ii) with an explicit look-up of local features(iii). None of these three components is new but the combination is very sound and the paper shows that it can improve the quality and generalization with this approach. This is in contrast to approaches which miss something, e.g. neglect camera geometry or use convenient grid representations that do not scale. The paper cites recent and highly related works sufficiently. The section can be improved by explaining the (dis)similarities between the local features used in Pix2Mesh and this work. Quality: The overall quality of this paper is good. The figures are helpful and qualitative examples give a good impression of the quality of the results and the outcome of specific experiments. I liked the experiments which explain the design choices in more detail (Sec. 4.3 and Fig. 8). Further, the experiments include comparisons between the GT camera pose and the estimated camera pose, which point out the dependency on the camera parameters of this approach. I miss a comparison with a naive baseline replacing the second decoder stream by adding to the SDF based on the projection to a foreground (add 0) or background pixel (add inf) in the image. While this baseline cannot add to the reconstructed volume it can generate holes, which would add detail. The quantitative evaluation uses common metrics such as IoU and CD; this is OK but does not do a good job on highlighting the advantages of this method, which is reconstruction of details. Consider using the F-score for the evaluation (see Tatarchenko et al. “What Do Single-view 3D Reconstruction Networks Learn?,” CVPR 2019). Clarity: The paper is well-written and gives enough information to implement the method. A minor issue are the statements about "infinite resolution"(l46) and "continuously sampling"(l209) could be misunderstood and written clearer to point out that this approach allows to freely sample the SDF. Significance: This work succeeds in improving the visual quality of single image reconstruction methods and the experiment with multiple views shows how this work can be extended and its ideas be used in related tasks. Quantitatively the work is on par with other related methods, which could be a limitation of the evaluation metrics. My biggest concern is the missing naive baseline, which would give us more information about the significance of the local feature decoder. Taking everything into account, I tend towards accepting this work. Questions: Is the signed distance approximately euclidean for the output of both streams? What does the signed distance function look like only for the local stream? Does it add only the details or does it contain the whole object? Minor mistakes: l.40: 'A' SDF l.58: To the best 'of' our ... l.147: grou'n'dtruth

Reviewer 2



To me, the main issue with this paper is that it tries to claim as a contribution the use of implicit function. Since at least 3 CVPR19 papers have already presented a similar idea (which is problematic to me, even if formally ok and not ground for rejection) I think this is a bad strategy for this paper, which has another contribution which I think is valid, useful and improve results, but is lost because of this presentation: it does not appear anywhere in the title and not clearly in the abstract. Simply looking at the title and abstract, I would not be likely to read the paper and would simply discard it as one more late paper building on the same idea. I would also have no chance to find it back if I wanted to cite it. Clearly not related work, but "PIFu: Pixel-Aligned Implicit Function for High-Resolution Clothed Human Digitization Shunsuke Saito*, Zeng Huang*, Ryota Natsume*, Shigeo Morishima, Angjoo Kanazawa, Hao Li" has a similar approach using local features, it might be nice to discuss the relation of the two works in the final version === Post-rebuttal please include the new dataset and experiments including more camera variation promised in the rebuttal

Reviewer 3



Positives: 1) I like the overall approach and the insight that when answering per-point queries, leveraging corresponding image-based features via reprojection should also help. Similar insights have been exploited in learning-based multi-view reconstruction works, but this approach is novel in context of single-view 3D reconstruction. 2) The paper is generally well written, easy to follow, and reports the desired ablations (though these do not necessarily support the claims, see below). Concerns/Comments/Questions: 1) My primary concern (and the main reason for leaning towards rejection) is that the central contribution of using 'local features' does not help empirically. I am judging this in the setting with 'estimated pose' (and not known pose, as this is additional information that is hard to acquire at inference). Judging by Table 3, the 'global network' is slightly better than the proposed approach ('Two stream, est') in IoU, similar in CD, and slightly worse in EMD. This shows that using the additional stream with local features (if camera pose is predicted) does not necessarily help. Similarly, the improvement of 'Ours_cam' over Occnet is only marginal. 2) I am concerned about certain aspects of camera prediction: a) How are symmetric objects handled e.g. if a table is square, how can the network predict the true camera pose. b) What is the variation in camera poses? From what I recall, the data from Choy et. al. always has a camera pointing towards origin, and a fixed elevation and cyclo-rotation, effectively having only 1 degree of freedom. If this is indeed the case, this should make the camera prediction simple, and I feel that this approach would degrade more in settings with larger camera variation (e.g. actual 6D freedom) compared to methods that do not use explicit camera prediction. It would really help the paper if experiments in settings with larger camera variation are shown. 3) Some additional comments (not central to the rating): a) I am currently evaluating the paper only in context of results in settings without known camera, and previous approaches do tackle reconstruction in settings with known camera e.g. Kar et. al., "Learnt Stereo Machines", and have a similar ideology of propagating image features to voxels, and this paper would then need to compare to these. I would also strongly recommend renaming 'Ours' to 'Ours + gt cam' and 'Ours_cam' to 'Ours', because currently method denoted as 'Ours' using extra information that baselines do not, and is not really tackling a 'Single-view 3D reconstruction' task as normally defined. b) I am curious why 'One stream' with 'ground-truth' is worse that with 'estimated'? ----- Overall, while the paper has an interesting central idea, the empirical results (in the setting with predicted pose) do not convince the reader that it is adds a significant additional value in terms of improving performance as the results (Table 3) are mixed, or at best indicate marginal improvement. ---------- Updates after rebuttal: I think the primary argument made in the rebuttal is that (in the setup with predicted camera) even though the quantitative results are only marginally better, the qualitative results are more impressive, and I think that is true. Additionally, the f-score metric reported shows slightly larger gains, and I'd overall be happy to increase my rating to marginal accept. That said, I really do hope the experiment with more camera variation would be added to the main paper as the authors promised in the rebuttal, as the current rebuttal experiment only shows that reprojection error is less, not that it helps in the downstream task.

[Author Response · NeurIPS 2019]

We thank the reviewers for the helpful feedback. All reviewers noted the novelty and superior performance of DISN, and the clarity of the exposition. We introduce a detail-preserving implicit surface network that generates high-quality 3D shapes by extracting local features. Although our quantitative results don't exceed existing methods by a large margin, we believe *the qualitative results (including 30 pages of figures in Supplementary)* are adequate to show that DISN achieves state-of-the-art performance on single-view reconstruction. We address major concerns raised by the reviewers.

**R1.Q1: F-Score Comparison** Please refer to Table 1 for F-Score results. We will add this experiment in our revision.

| Threshold(%) | 0.5% | 1% | 2% | 5% | 10% | 20% |
|---|---|---|---|---|---|---|
| 3DCNN | 0.064 | 0.295 | 0.691 | 0.935 | 0.984 | 0.997 |
| IMNet | 0.063 | 0.286 | 0.673 | 0.922 | 0.977 | 0.995 |
| DISN gt cam | 0.079 | 0.327 | 0.718 | 0.943 | 0.984 | 0.996 |
| DISN est cam | 0.070 | 0.307 | 0.700 | 0.940 | 0.986 | 0.998 |

Table 1: F-Score for varying thresholds (% of reconstruction volume side length, same as "What Do Single-view 3D Reconstruction Networks Learn?"(Tatarchenko et al, CVPR 2019)) on all categories.

**R1.Q2: Comparison with a baseline replacing local feature extraction with background subtraction** Please refer to Figure 1. Compare to results from global features, the baseline can generate some holes. However, it suffers from inaccurate reprojection and noncontinuous SDF prediction.

**R1.Q3: What global and local stream's sdf look like** As shown in Figure 1, the global branch produces the overall shape and the local branch adjusts the sdf based on local details.

Figure 1: Baseline is as suggested by R1. We also show the reconstruction of DISN's global branch and the chair back details generated by DISN's local branch.

**R1.Q4: Relations to Pix2Mesh Local Feature Module** Pixel2mesh reconstructs 3D model by deforming an ellipsoid, which makes them impossible to produce different topology and generate different shape details. However, by taking advantage of predicting implicit surface, DISN is able to generate shape details according to local features.

**R2.Q1: Statement of contributions is misleading** Thanks for pointing this out. We will further emphasize local feature extraction as our main contribution in our revision.

**R2.Q2: Add citations for the related voxel and point cloud methods** We will conduct a more comprehensive literature review and also add the citations.

Figure 2: Green dots are sampled ground truth points. Red dots are projected points using estimated camera parameters. The lamp has an offset in 3d space.

**R2.Q3: Relations to "PIFu"** Thanks for suggesting this related work which was released at the same time as NeurIPS 2019 submission deadline. We will discuss the relations of our paper and this paper in our final version.

**R3.Q1: Quantitative results are not superior to previous methods by a large margin.** 1) Although DISN doesn't outperform 'global' by a large margin quantitatively, the qualitative results illustrates that DISN is able to reconstruct various shape details that all methods without local feature extraction fail to generate as acknowledged by other reviewers. To the best of our knowledge, DISN is the first work that is able to generate fine-grained details in 3D shapes (such as holes in the rifle in our teaser). Moreover, we also provide a wide range of qualitative results in Supplementary, which sufficiently show our superior performance compared to state-of-the-art methods. 2) "What Do Single-view 3D Reconstruction Networks Learn?"(Tatarchenko et al, CVPR 2019) shows that the qualitative results are not necessarily related to the quality of generated shape details. Please also refer to R1.Q1 for F-score comparison where DISN constantly outperforms the state-of-the-art methods. In Figure 1 we also show the importance of our local feature extraction to the detail reconstruction.

**R3.Q2a: Camera prediction for symmetric objects** We show some examples of camera prediction of symmetric objects in Figure 2 that have large projection errors in 3D but small errors on 2D. 1) In most cases, incorrect camera prediction due to symmetric ambiguity has no impact on 2d reprojection, therefore no impact on local features query. 2) Even the 2d reprojected location has a shift of 2.95 pixels on average (Table 2 in our paper), this shift will be decreased when we query local features on higher-level feature maps with smaller dimensions.

**R3.Q2b: Small Camera variance of Choy's Rendering Dataset** Camera prediction is not our main contribution. To make a fair comparison, we use the same dataset as previous methods. In Choy's dataset, the distance and field of view are fixed instead of elevation and cyclorotation. The degree of freedom is 3 instead of 1. As suggested by the reviewer, we enlarge the camera variation by making the camera not to point towards the origin. We simply change $C_x, C_y$ in intrinsic matrix from fixed to random numbers in $(-40, 40)$ and trained a new camera prediction network by predicting both extrinsic and intrinsic parameters. The reprojection error for this new setting is $3.54$, while the reprojection error for "Choy" dataset is $2.95$ (Table 2 in our paper). Therefore, our camera prediction method is robust to larger variation and higher DOF. We are also preparing a new dataset for 3D reconstruction by rendering with greater camera variation. We will train a new camera prediction network and add this experiment to our final version.

**R3.Q3a: Confusing notations.** We apologize for the confusion and will revise in the final version.

**R3.Q3b: One stream with gt camera is worse than estimated.** The ground truth and the estimated results should be swapped. We will correct the typo in the final version.

[Meta-Review · NeurIPS 2019]

Following the authors' rebuttal, there is a consensus to accept the paper. The authors are requested to address the reviewers' concerns and integrate additional material into the revision, as discussed in the rebuttal.